# Therapeutic effect of Sanhua decoction on rats with middle cerebral artery occlusion and the associated changes in gut microbiota and short-chain fatty acids

Yiming Ni[1,2]☯, Liangyin Cai[3]☯, Xiaojun Gou[4]☯, Wenjie Li[2], Mingmei Zhou📧[1‡]*, Ying Huang[2‡]*

1 Institute of Interdisciplinary Integrative Medicine Research, Shanghai University of Traditional Chinese Medicine, Shanghai, China, 2 Experimental Research Center, China Academy of Chinese Medical Sciences, Beijing Key Laboratory of Research of Chinese Medicine on Prevention and Treatment for Major Diseases, Beijing, China, 3 Department of Pharmacy, Zhongshan Hospital Wusong Branch, Fudan University, Shanghai, China, 4 Central Laboratory, Baoshan District Hospital of Integrated Traditional Chinese and Western Medicine of Shanghai, Shanghai University of Traditional Chinese Medicine, Shanghai, China

☯ These authors contributed equally to this work.
‡ MZ and YH also contributed equally to this work and should be considered corresponding authors.
* zhoumm368@163.com (MZ); huangying0518@126.com (YH)

## Abstract

Sanhua decoction (SHD), a traditional prescription, has long been used in treating ischemic stroke (IS). However, the therapeutic effect of SHD and the associated changes in gut microbiota and short-chain fatty acids (SCFAs) are uncertain. In this study, a rat model of IS was established by the middle cerebral artery occlusion (MCAO). By evaluating the cerebral infarct area and brain tissue pathology, it was found that SHD ameliorated IS-related symptoms in MCAO rats. Using 16S rRNA gene sequencing, we found that SHD reduced abnormally elevated *Lactobacillus* and opportunistic pathogens such as *Desulfovibrio*, but increased some beneficial bacteria that produce SCFAs, including *Clostridia*, *Lachnospiraceae*, *Ruminococcaceae*, and *Coprococcus*. KEGG analysis revealed that SHD regulates several pathways, including D-arginine and D-ornithine metabolism, polyketide sugar unit biosynthesis, and cyanoamino acid metabolism, which are significantly altered in MCAO rats. By gas chromatography-mass spectrometry detection of SCFAs, we found that fecal acetic acid, valeric acid, and caproic acid were significantly increased in MCAO rats, whereas propionic acid and isobutyric acid were decreased. SHD reversed the changes in acetic acid and propionic acid in the model rats and significantly increased fecal butyric acid. In addition, MCAO rats had significantly higher serum levels of acetic acid, butyric acid, isovaleric acid, and valeric acid, and lower levels of caproic acid. Altered serum levels of butyric acid, isovaleric acid, valeric acid, and caproic acid were restored, and the level of isobutyric acid was reduced after SHD administration. Spearman analysis revealed that cerebral infarct area had a strong correlation with *Bifidobacterium*, *Desulfovibrio*, *Lachnospiraceae*, *Lactobacillus*, acetic acid, valeric acid, and caproic acid. Overall, this study demonstrates for the first time that the

**Data Availability Statement:** The datasets presented in this study can be found in online repositories. The names of the repository and accession number can be found: SRA, PRJNA875206 (https://www.ncbi.nlm.nih.gov/sra/PRJNA875206).

**Funding:** This study was financially supported by the Fundamental Research Funds for the Central Public Welfare Research Institutes (JJPY2022024). The funders had no role in study design, data collection and analysis, decision to publish, or preparation of the manuscript.

**Competing interests:** The authors have declared that no competing interests exist.

**Abbreviations:** SHD, Sanhua decoction; IS, ischemic stroke; MCAO, middle cerebral artery occlusion; SCFAs, short-chain fatty acids; BBB, blood-brain barrier; TTC, 2,3,5-triphenyl tetrazolium chloride; GC-MS, gas chromatography-mass spectrometry; Ctrl, control; KEGG, Kyoto Encyclopedia of Genes and Genomes; HPA, hypothalamic-pituitary-adrenal; CCA, common carotid artery; ICA, internal carotid artery; ECA, external carotid artery; ASVs, amplicon sequence variants.

effect of SHD on IS may be related to gut microbiota and SCFAs, providing a potential scientific explanation for the ameliorative effect of SHD on IS.

## Introduction

Ischemic stroke (IS) is a leading cause of death and disability worldwide [1] and its primary lesion is a cerebral infarction. As a result of inadequate blood supply to cerebral tissue, there is an initial reversible loss of tissue function, followed by a possible infarct with loss of neurons and supporting structures [2]. The main factors causing IS include inflammation, oxidative damage, energy metabolism dysfunction, excitotoxicity, $Ca^{2+}$ overload, apoptosis, and autophagy [3], and its common comorbidities are hyperhomocysteinemia, hypertension, diabetes mellitus, and hyperlipidemia [4, 5].

Sanhua decoction (SHD), a famous ancient Chinese herbal prescription recorded in Suwen Bingji Qiyi Baomingji (a classic of traditional Chinese medicine dating back to 1188 AD in the Jin Dynasty), has been widely used clinically to treat IS due to its special advantages on multiple targets [6]. Based on the theory of traditional Chinese medicine, SHD can soothe Qi flow and unclog sweat pores. Clinically, SHD can play a role in reducing blood viscosity and regulating circulatory disorders [7]. SHD consists of 4 herbs, namely Radix et Rhizoma Rhei *(Dahuang)*, Rhizoma et Radix Notopterygii *(Qianghuo)*, Fructus Aurantii Immaturus *(Zhishi)*, and Cortex Magnolia officinalis *(Houpu)* [8]. Modern experimental studies show that SHD modulates PI3K/Akt/CREB1 and TNF pathways and targets aquaporin 4 to protect against IS [7, 9].

The bidirectional interaction between brain and gut is a well-known mechanism in both healthy and diseased states [10]. Stroke affects intestinal permeability, thereby inducing the gut microbiota to become more toxic [11]. As IS is associated with more opportunistic pathogens, such as *Enterobacter*, *Megasphaera* and *Oscillibacter* [12, 13], the specific role of the microbiota in IS deserves attention. However, the mechanisms by which the microbiota influence the host remain unclear [14]. If modulation of the gut microbiota is beneficial in IS, individuals who lack a healthy or anti-inflammatory gut microbiota at the time of stroke may benefit from the gut microbiota-focused therapeutic strategy, such as probiotics, prebiotics, and fecal microbiota transplantation [11, 15]. SHD mainly contains flavonoids (in Fructus Aurantii Immaturus and Radix et Rhizoma Rhei), anthraquinones (in Radix et Rhizoma Rhei), coumarins (in Rhizoma et Radix Notopterygii), phenylpropanoid glycosides, alkaloids and lignans (in Cortex Magnolia officinalis) [8], which have been shown to be closely related to the gut microbiota [16].

Short-chain fatty acids (SCFAs), derived from bacteria-dependent fiber hydrolysis, are key signaling molecules for gut-brain communication. After entering the systemic blood and crossing the blood-brain barrier (BBB) into the brain, SCFAs could modulate the function of the BBB, astrocytes, microglia, and neurons, and affect brain function in development, health, and disease [17–19]. Previous studies have suggested that the abnormalities in SCFAs would increase the susceptibility to stroke and affect prognosis, while SCFAs supplementation could prevent stroke through immune mechanisms [1, 11, 12, 20, 21]. As SCFAs are metabolites of gut microbiota, the transplantation of SCFA-rich fecal bacteria has been shown to be effective for IS [22].

Based on the above, that SHD, the traditional Chinese medicine prescription has been proven effective for IS, most of its ingredients are closely related to gut microbiota, and the gut microbiota metabolites, SCFAs are beneficial for IS. It is worth exploring the effect of SHD on gut microbiota, and whether it can affect IS through SCFAs, the IS-related gut microbiota

metabolites. In this study, a rat model of IS was established by middle cerebral artery occlusion (MCAO). The successful MCAO model was confirmed by 2,3,5-triphenyl tetrazolium chloride (TTC) staining and hematoxylin-eosin (HE) staining. We investigated the effect of SHD on gut microbiota and SCFAs on the IS rat model by 16s rRNA gene sequencing and gas chromatography-mass spectrometry (GC-MS)-based metabolomics, and further clarified the relationship between cerebral infarction, gut microbiota, and SCFAs to summarize the mechanism of SHD on IS, and it is hoped to provide reference for further research.

## Methods and materials

### Preparation of SHD

SHD consists of 4 traditional Chinese herbs, namely Radix et Rhizoma Rhei *(Dahuang)*, Rhizoma et Radix Notopterygii *(Qianghuo)*, Fructus Aurantii Immaturus *(Zhishi)*, and Cortex Magnolia officinalis *(Houpu)*. All the traditional Chinese medicine decoction slices were purchased from Beijing Tong Ren Tang Co., Ltd. (Beijing, China) and authenticated by Dr. Xirong He from China Academy of Chinese Medical Sciences. In a ratio of 1:1:1:1, 4 Chinese herbal slices were soaked in 10 times the amount of water for 30 minutes and then decocted with water for 30 minutes. The decoction was filtered at room temperature and concentrated to 1 g/mL stock for future use.

### Animal experimental design

A total of 24 male Sprague-Dawley rats weighing between 230 and 250 g were purchased from Beijing Huafukang Biotechnology Co., Ltd. (SCXK (Beijing) 2019–0008). All experimental procedures were approved by the Experimental Research center of China Academy of Chinese Medical Sciences (Number: ERCCACMS21-2307-03). All animal experiments and care were performed by investigators trained in animal care. Rats were randomly assigned to the control (Ctrl, n = 6), MCAO (n = 12) and Sham (n = 6) groups and were housed under a 12-hour light/dark cycle with free access to food and water at a temperature of 23±2˚C.

The MCAO rat model was established by the suture-occluded method according to our previous methods [23]. After anesthesia with 1% pentobarbital (40 mg/kg BW, *i.p.*), the right common carotid artery (CCA), internal carotid artery (ICA), and external carotid artery (ECA) of MCAO rats were exposed and isolated. The CCA and ECA were ligated 5 mm from the bifurcation of the ECA and ICA and at the end of the CCA, respectively, and the preloaded suture was pulled to occlude the ICA blood flow. A small incision was then made at the bifurcation of the ICA and CCA and the monofilament was inserted. The suture was released and advanced approximately 18–20 mm until resistance was encountered, then the ICA was ligated, leaving the suture in place to prevent the monofilament from falling out. After 90 min post ischemia, the monofilament was gently removed from the ICA to perform reperfusion (**Fig 1A**). Different from the MCAO group, the monofilament in the Sham group was inserted only 10 mm into the ICA, not into the MCA. The wound was disinfected with iodine and sutured.

After MCAO surgery, the rats were allowed to recover on a heating mat (37˚C) until awakening, and were randomly divided into Model (n = 6) and SHD (n = 6) groups. SHD rats were administered SHD intragastrically at 7.2 g/kg body weight for ten consecutive days. The Ctrl, Sham, and Model rats were given with the same volume of purified water (**Fig 1B**). Meanwhile, the rats' appetite, body temperature, mobility, and health were monitored daily. When rats were near death, they were immediately euthanized by injection of sodium pentobarbital.

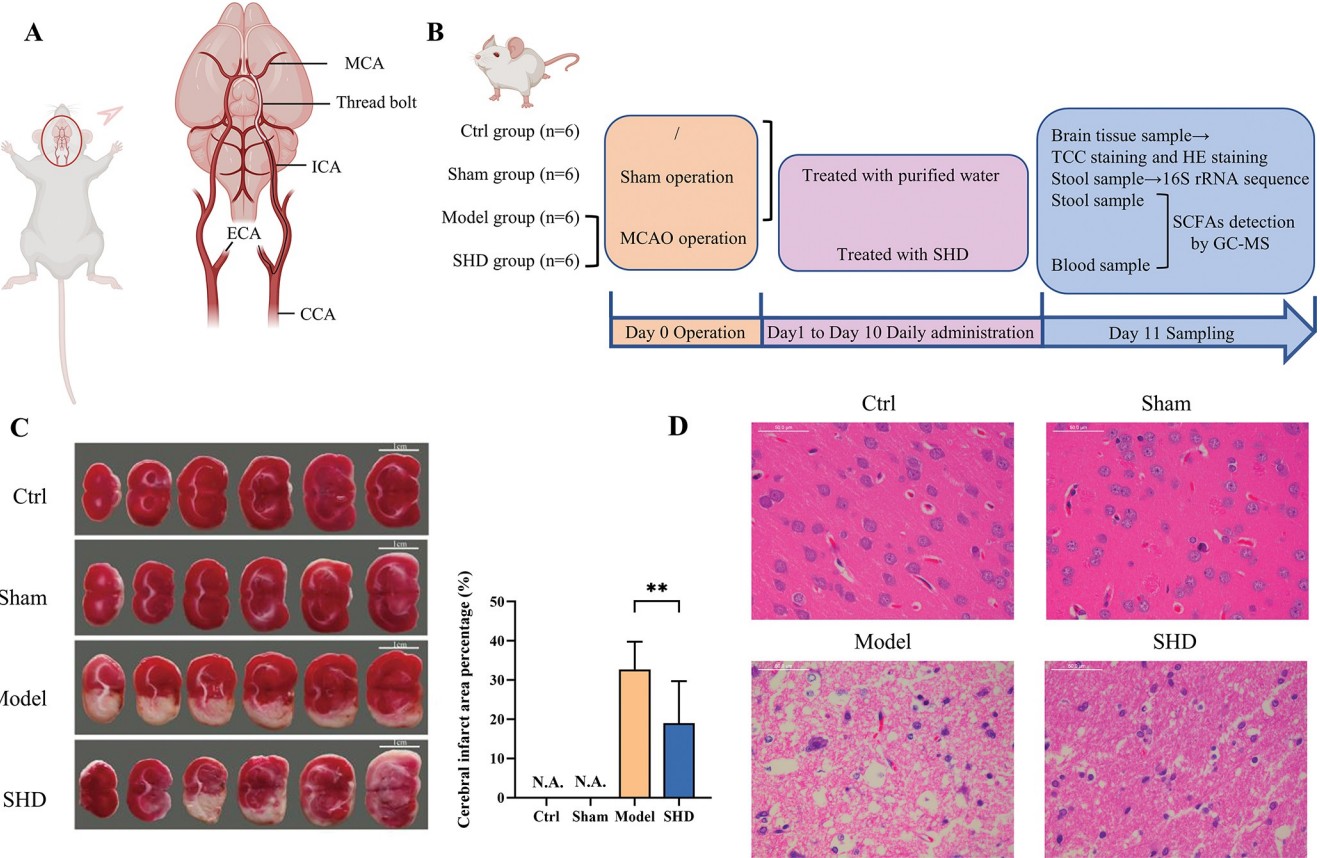

**Fig 1. Experimental design for MCAO rats and pathological results in rats.** (A) The operation of the MCAO model. The right CCA, ICA, and ECA were exposed and isolated. The CCA and ECA were ligated 5 mm from the bifurcation of the ECA and ICA and at the end of the CCA, respectively, and the preloaded suture was pulled to obstruct the ICA blood flow. A small incision was then made at the bifurcation of the ICA and CCA and the monofilament was inserted. The suture was loosened and advanced approximately 18–20 mm until resistance occurred, then the ICA was ligated, leaving the suture in place to prevent the monofilament from falling out. After 90 min post ischemia, the monofilament was gently removed from the ICA to perform reperfusion. (B) Design of the animal experiment. After the MCAO model was established, the Ctrl, Sham, and Model groups were fed with purified water, and SHD was administered with SHD for 10 consecutive days. On day 11, samples were taken after euthanasia. (C) TTC-stained sections of brain tissue and percentage of cerebral infarct area from the four groups of rats (**** $P < 0.0001$). (D) HE staining of brain tissues (200×). Abbreviations: MCAO, middle cerebral artery occlusion; CCA, common carotid artery; ICA, internal carotid artery; ECA, external carotid artery.

## Pathomorphology of brain

Rats were anesthetized with 1% pentobarbital (40 mg/kg body weight) by intraperitoneal injection 24 h after the last administration of the drug. The brain was isolated after craniotomy. MCAO-induced cerebral infarction was confirmed by TTC staining. Briefly, the brain tissue was cut into six posterior coronal slices before incubation with 2% TTC. The percentage of cerebral infarct area was calculated by using the Image J1.41 software. For HE staining, the brain tissues were fixed in 4% paraformaldehyde, routinely dehydrated, embedded in paraffin, sectioned, stained with HE, mounted, and observed under a Leica light microscope.

## 16S rRNA gene sequence

All feces were collected in sterile containers the day after final dosing and stored in -80˚C freezers until processing. Total bacterial DNA was extracted using the OMEGA Soil DNA Kit (M5635-02) (Omega Bio-Tek, USA), and the quantity and quality of extracted DNA were further measured using the NanoDrop NC2000 spectrophotometer (Thermo Fisher Scientific,

USA) and agarose gel electrophoresis, respectively. Bacterial 16S rRNA genes in the V3-V4 region were sequenced using primers 338F (5′-ACTCCTACGGGAGGCAGCA-3′) and 806R (5′-GGACTACHVGGGTWTCTAAT-3′) on the Illumina NovaSeq platform using the Nova-Seq 6000 SP Reagent Kit (500 cycles).

Microbiome bioinformatics was performed using QIIME2 2019.4. Raw sequence data was demultiplexed using the demux plugin. The DADA2 plugin was then used for quality filtering, denoising, merging and chimera removal. Non-singleton amplicon sequence variants (ASVs) were aligned using MAFFT and a phylogeny was constructed using FastTree 2. Alpha (Chao1, Simpson, Shannon, Pielou's evenness, Observed species, Faith's PD, and Good's Coverage indices) and beta diversity metrics (principal coordinate analysis and non-metric multidimensional scaling were performed based on unweighted UniFrac distances and Bray-Curtis metrics, respectively) were estimated using the diversity plugin. The classify sklearn naive Bayes taxonomy classifier in the feature classifier plugin was used against the SILVA Release 132 database to assign the taxonomy.

## GC-MS analysis

Extraction and quantification of SCFAs were performed according to previous reports [24, 25]. To extract fecal SCFAs, 100 μL of 125 μg/mL alien acid, 50 μL of 15% phosphoric acid, and 400 μL of diethyl ether were added alternately to aliquots of fecal samples in centrifuge tubes. To extract serum SCFAs, 10 μL of 75 μg/mL alien acid, 50 μL of 15% phosphoric acid, and 140 μL of diethyl ether were added alternately to aliquots of serum samples. The mixtures were vortexed for 1 min and centrifuged at 12000×g for 10 min at 4˚C, and the supernatant was used for GC-MS analysis. The standard curve method was used to determine the concentrations of the SCFAs. GC-MS (Thermo Trace 1310-ISQ LT) was applied with an Agilent HP-INNOWAX capillary column (30 m×0.25 mm, i.d. 0.25 μm), and the sample volume for analysis was set at 1 μL. Helium, the carrier gas, was set at a flow rate of 1.0 mL/minute. Oven temperatures were programmed from 90˚C to 120˚C at 10˚C/minute, to 150˚C at 5˚C/minute, and to 250˚C at 25˚C/minute, with a 2-minute hold. In addition, the injection port, transfer line, and ion source temperatures were set to 250˚C, 250˚C, and 300˚C, respectively. An electron ionization source and selective ion monitoring were used. The electronic impact was at 70 eV.

## Statistical analysis

GraphPad Prism (v. 8.0) software was used for statistical analysis. All data are presented as the mean ± standard deviation. The Shapiro-Wilk normality test was used to test for normality. One-way ANOVA followed by post hoc Tukey's test or Dunn's test was used to analyze the statistics of four groups. Spearman's correlation analysis was used to determine the correlations between the levels of the different analytes. Significant differences were indicated as follows: * $P < 0.05$, ** $P < 0.01$, *** $P < 0.001$, and **** $P < 0.0001$.

## Results

### SHD improved the pathological brain morphology in MCAO rats

To evaluate the efficacy of the SHD in the treatment of IS, TTC staining and HE staining of brain tissues were examined. The TTC staining results show that the brain tissue sections of the Ctrl and Sham groups were completely stained red, whereas the brain tissues of the rats in the MCAO group showed an extensive pale staining, and the percentage of cerebral infarction reached 32.71%±6.43%, indicating the successful establishment of the MCAO model, which

could accurately simulate the disease state of IS. The brain tissues of SHD-treated rats showed a state of recovery. SHD supplementation significantly reduced the size of cerebral infarction compared with the model group (Fig 1C). In addition, the pathological morphology of HE staining was performed on brain tissues to validate the beneficial effects of SHD on MCAO rats. Brain cellular structures in the Ctrl and Sham groups were intact, cells were neatly arranged, nuclei were centered, and nucleoli were visible. Brain tissue sections from the Model rats showed loose cell structure, neuronal degeneration, necrosis, pyknosis, and disappearance of nuclei, indicating extensive brain tissue damage. After 10 days of SHD treatment, the brain tissue of the rats was restored in terms of reduced necrotic lesions, increased cell number, completed cell structures, and an ordered cell arrangement (Fig 1D). The result indicated that SHD could significantly reduce ischemic necrosis areas in brain tissue and alleviate pathological injury.

## SHD changed alpha and beta diversity of gut microbiota in MCAO rats

A relationship between gut microbiota and IS is increasingly supported. However, potential predictors of IS are rarely analyzed [26]. Therefore, we investigated the influence of SHD on gut microbiota using 16S rRNA sequencing analyses. The ASV-level alpha diversity indices, which evaluate the richness, diversity, evolutionary diversity, uniformity, and coverage of gut microbiota were calculated to observe significant differences among four groups of rats. The Chao1 ($P$ = 0.0110), Simpson ($P$ = 0.0016), Shannon ($P$ = 0.0010), Pielou's evenness ($P$ = 0.0007), Observed species (0.0100), Faith's PD ($P$ = 0.0275) indices of the Model group were significantly decreased compared with the Ctrl group. SHD administration could restore the diversity, although the indices did not show statistical differences (Fig 2A). The Good's Coverage index showed no statistical difference among the Ctrl, Model, and SHD groups.

The effect of SHD treatment on the β-diversity of gut microbiota was investigated using principal coordinate analysis (Fig 2B) and nonmetric multidimensional scaling (Fig 2C) plots based on ASVs, which visualized the structural differences of microbiota among samples. Our results showed that the gut microbiota in the Ctrl, Model, and SHD groups were distinguishable, making it clear that SHD treatment significantly altered the gut microbiota composition of rats with IS.

## SHD altered gut microbiota composition in MCAO rats

Venn diagram and histogram were used to visualize the shared and unique ASVs among the four groups (Fig 3A). The taxonomic compositions and abundances showed that *Firmicutes* and *Bacteroidetes* were dominant at the phylum level among the four groups. At the genus level, the genera *Lactobacillus*, *Prevotella*, *Bacteroides*, *Ruminococcus*, *Enterococcus*, *Blautia*, *Oscillospira*, and *Shigella* constituted the majority of the gut microbiota (Fig 3B). Furthermore, the linear discriminant analysis effect size was applied to analyze differentially abundant taxa. At the genus level, the Ctrl group was dominated by *Prevotella*, *Ruminococcus*, *Blautia*, *Roseburia*, *Dorea*, *SMB53*, *[Eubacterium]*, *Sutteralla*, *Clostridium*, *Staphylococcus*, *[Ruminococcus]*, and *Barnesiella*. *Bifidobacterium*, *Lactobacillus*, and *Desulfovibrio* were the three major genera in the Model group. In the SHD group, *Allobaculum*, *Phascolarctobacterium*, *[Prevotella]*, *Streptococcus*, *Clostridium*, *Coprococcus*, *Turicibacter*, *Veillonella*, and *rc4_4 were dominant* (Fig 3C).

According to the above results, we compared the relative abundance of some specific bacteria. Some recent studies reported that *Enterobacteriaceae*, *Bifidobacterium*, *Lactobacillus*, and *Desulfovibrio* were enriched in higher-risk stroke patients [27, 28]. However, the levels of *Enterobacteriaceae* ($P$ > 0.05, Fig 4A) and *Desulfovibrio* ($P$ > 0.05, Fig 4B) did not showe

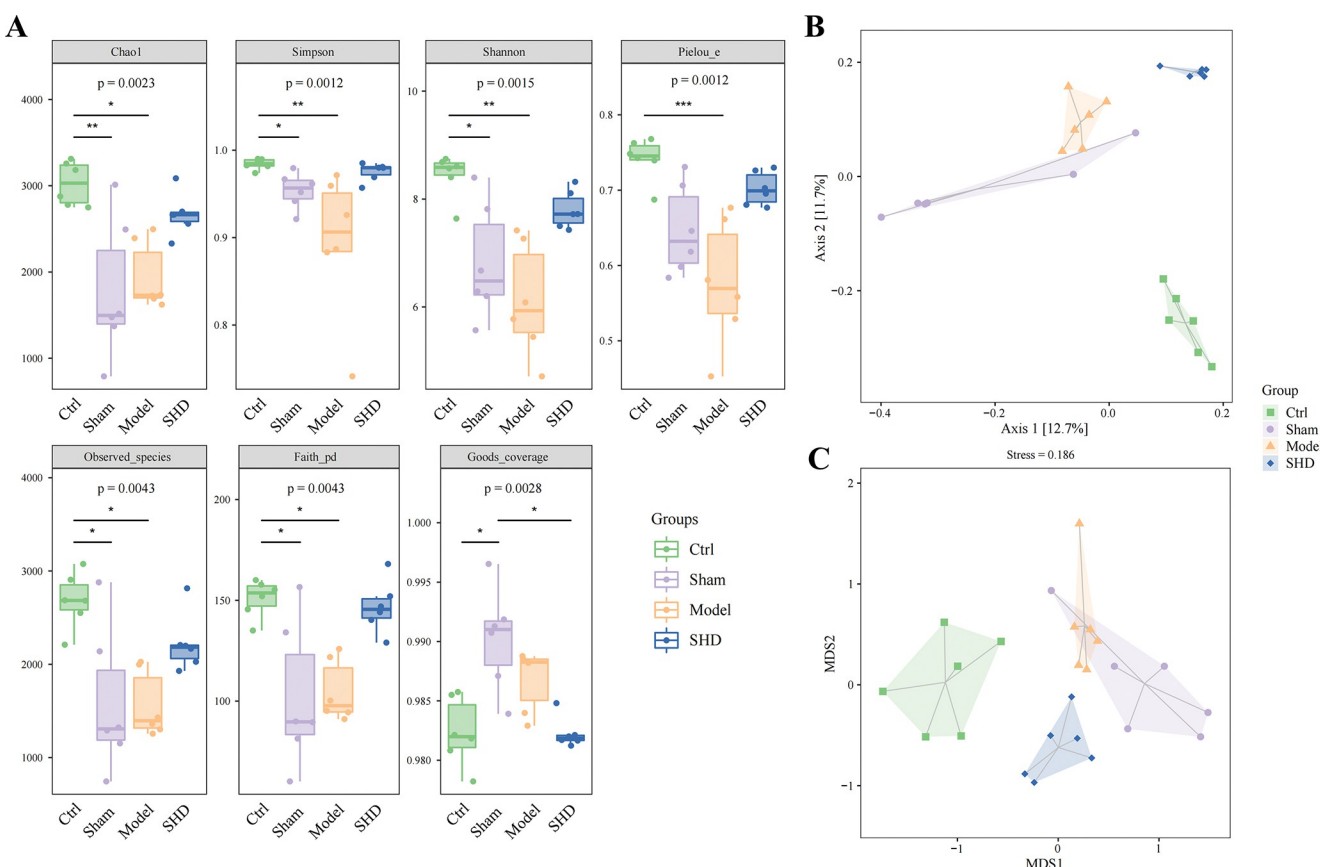

**Fig 2. Alpha and beta diversity of gut microbiota in MCAO rats.** (A) Comparison of alpha diversity between the gut microbiota of four groups of rats. Seven indices were used to evaluate the alpha diversity (Chao1, Observed species, Shannon, Simpson, Faith's PD, Pielou's evenness, and Good's coverage). (B) Unweighted UniFrac-based principal coordinate analysis showing the gut microbiota composition among the Ctrl, Sham, Model, and SHD groups. (C) Nonmetric multidimensional scaling based on Bray-Curtis metrics showing the gut microbiota composition among the Ctrl, Sham, Model, and SHD groups (n = 6). Statistical significance was considered at * $P < 0.05$, ** $P < 0.01$, and *** $P < 0.001$.

statistical differences between the Ctrl and Model groups, but the levels of *Lactobacillus* ($P < 0.0001$, **Fig 4C**) and *Bifidobacterium* ($P = 0.0358$, **Fig 4D**) were increased in the Model group compared with the Ctrl group. We also found significant levels of *Desulfovibrio* ($P = 0.0461$) and *Lactobacillus* ($P = 0.0006$) in MCAO rats after SHD administration. As propionic acid-producing microbiota and major gut colonizers in early life, *Clostridia* showed a decrease in the Model group ($P < 0.0001$) and were restored after SHD administration ($P < 0.0001$, **Fig 4E**). Moreover, members of the butyric acid producers, *Lachnospiraceae* and *Ruminococcaceae*, received the most attention due to their high abundance in the human colon [29]. They were found to be less abundant in high-risk stroke patients [27]. Therefore, we focused on analyzing the abundances of *Lachnospiraceae* and *Ruminococcaceae* and found that they were significantly depleted in the Model group ($P = 0.0005$ and $P = 0.0319$, respectively). However, the levels of *Lachnospiraceae* ($P = 0.0256$, **Fig 4F**) and *Ruminococcaceae* ($P = 0.0038$, **Fig 4G**) recovered after SHD treatment. SHD treatment also increased the abundance of *Coprococcus* ($P = 0.0107$, **Fig 4H**), which could also produce butyric acid [13, 19]. These results suggested that MCAO modeling led to an alternation and a compositional change of the resident gut microbiota. However, SHD administration could reduce the levels of opportunistic pathogens and generate some SCFAs-producing beneficial bacteria.

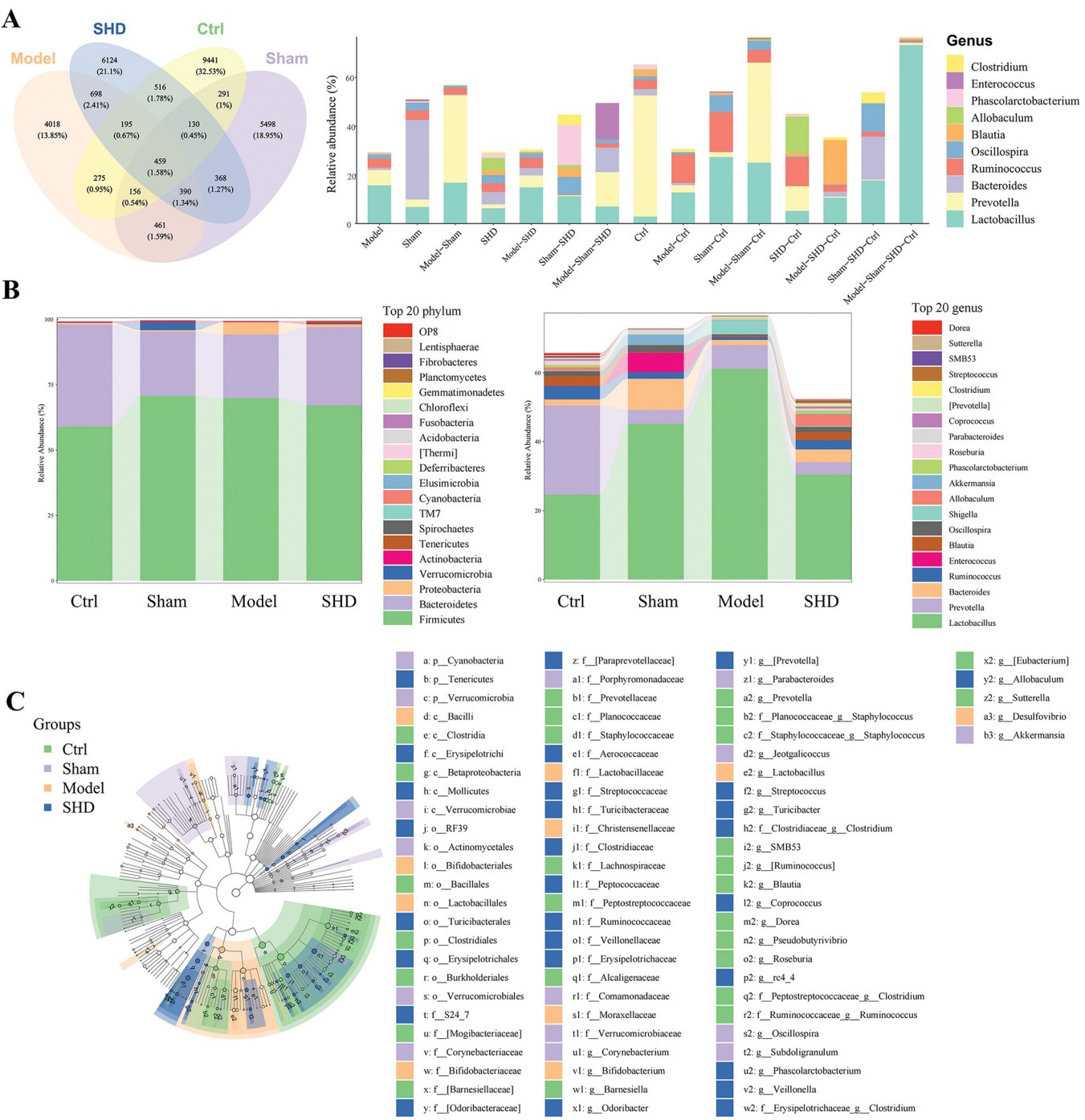

**Fig 3. Altered gut microbiota composition in MCAO rats.** (A) Venn diagram for ASVs and the abundance of ASVs at the phylum and genus level corresponding to each region in the Venn diagram. (B) Relative abundance of gut microbial taxa in feces at the phylum and genus level. (C) The taxonomic cladogram of LEfSe analysis (significance levels for LEfSe were $P < 0.05$ and linear discriminant analysis $> 2.0$).

## SHD altered gut microbial functions in MCAO rats

Gut microbiota affect host metabolic pathways, which may subsequently influence the neural network of the ischemic brain [13]. Phylogenetic Investigation of Communities by Reconstruction of Unobserved States 2 (PICRUSt2) was used to predict microbial functions.

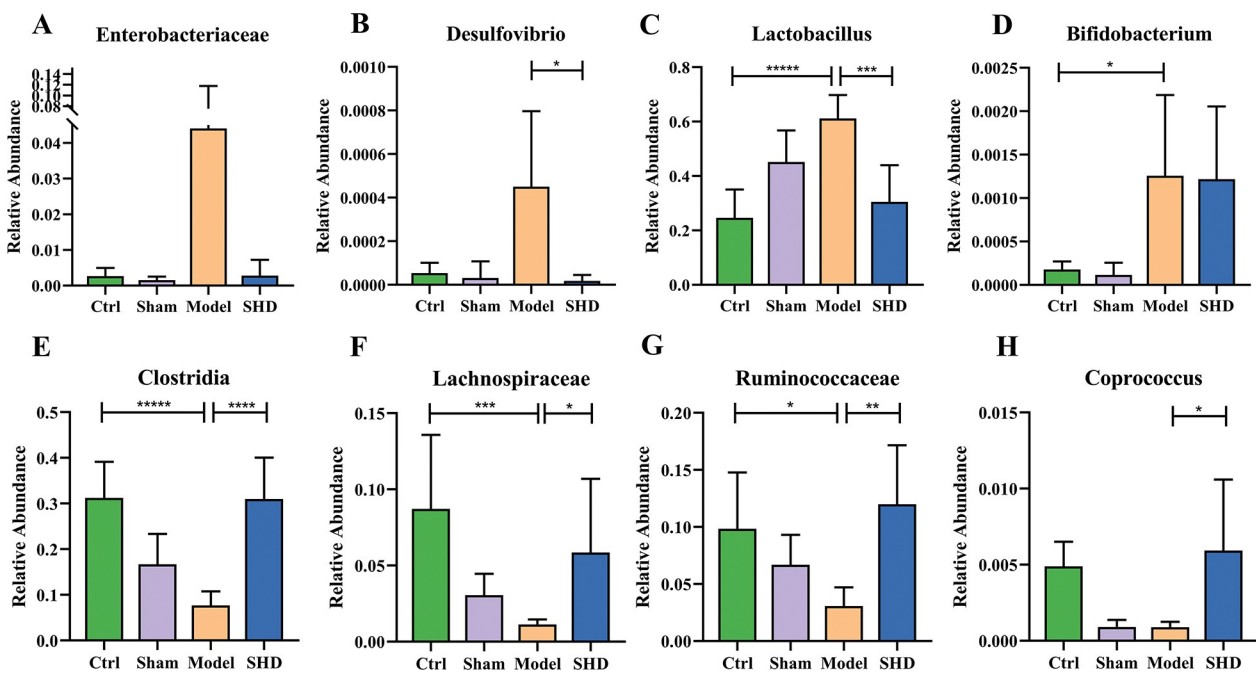

**Fig 4. Relative abundance of gut microbiota.** (A) *Enterobacteriaceae;* (B) *Desulfovibrio;* (C) *Lactobacillus;* (D) *Bifidobacterium;* (E) *Clostridia;* (F) *Lachnospiraceae;* (G) *Ruminococcaceae*; (H) *Coprococcus* (n = 6). Statistical significance was considered at * $P < 0.05$, ** $P < 0.01$, *** $P < 0.001$, and **** $P < 0.0001$.

Secondary functional pathways were obtained from the Kyoto Encyclopedia of Genes and Genomes (KEGG) database (https://www.kegg.jp/). Next, we examined the differential pathways between Ctrl and Model groups (**Fig 5A**) and between Model and SHD groups (**Fig 5B**) by analyzing KEGG metabolic pathways based on MetagenomeSeq. We found that compared with the Ctrl group, pathways including polyketide sugar unit biosynthesis and cyanoamino

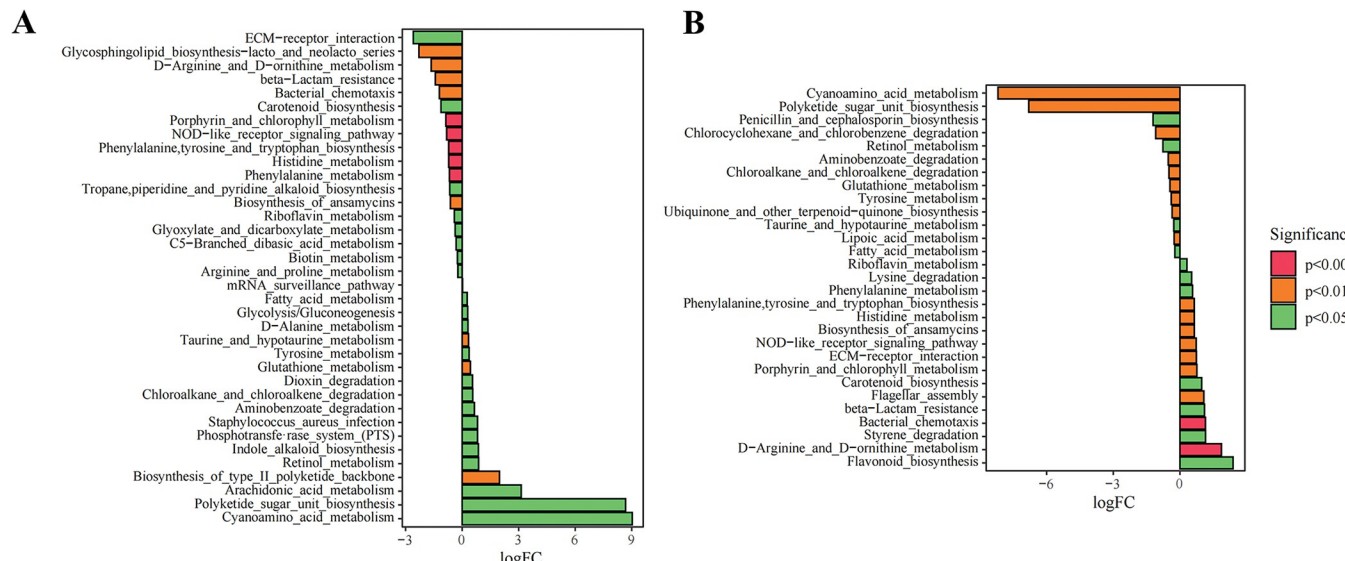

**Fig 5. Function prediction of microbial genes by PICRUSt2 analysis.** (A) Differential pathways between Ctrl and Model on KEGG (logFC > 0, pathways were upregulated by Model). (B) Differential pathways between Model and SHD on KEGG (logFC > 0, pathways were upregulated by SHD).

acid metabolism were up-regulated, and pathways including ECM-receptor interaction, glyco-sphingolipid biosynthesis lacto and neolacto series, and D-arginine and D-ornithine metabolism were down-regulated in the Model group. Interestingly, polyketide sugar unit biosynthesis, cyanoamino acid metabolism, and D-arginine and D-ornithine metabolism were restored in SHD-treated rats. These results showed that the altered gut microbiota by SHD induced changes in metabolic pathways, which may be the underlying mechanism of SHD in the treatment of IS.

## SHD selectively restored fecal and serum SCFAs in MCAO rats

Considering that SCFAs are generated by the metabolism of indigestible dietary fiber by the gut microbiota in the intestinal tract [11], and that some SCFAs-producing bacteria were significantly changed after SHD administration, we detected the SCFAs concentrations in feces and serum using a targeted metabolomics assay by gas chromatography-mass spectrometry. Three-dimensional principal component analysis plots were displayed to illustrate the overall variation between the Ctrl and Model groups with SHD intervention (**Figs 6A** and **7A**). The plots showed a clear separation of the Ctrl, Sham, and SHD groups from the Model groups. The heat map revealed the ability of SHD to alter SCFAs concentrations (**Figs 6B** and **7B**).

Total SCFAs were the sum of acetic acid, propionic acid, butyric acid, valeric acid, caproic acid, isobutyric acid, and isovaleric acid. The concentrations of total SCFAs in feces ($P = 0.0030$, **Fig 6C**) and serum ($P = 0.0003$, **Fig 7C**) were elevated in the Model group compared to those of the Ctrl group. However, SHD did not alter total SCFA levels in either serum or plasma samples ($P > 0.05$).

Compared with the Ctrl group, fecal acetic acid ($P = 0.0006$, **Fig 6D**), valeric acid ($P = 0.0330$, **Fig 6I**), and caproic acid ($P = 0.0076$, **Fig 6J**) were significantly increased in the Model group, while the levels of propionic acid ($P = 0.0087$, **Fig 6E**) and isobutyric acid ($P = 0.0238$, **Fig 6F**) were decreased. SHD administration selectively restored the levels of acetic acid ($P = 0.0354$, **Fig 6D**) and propionic acid ($P = 0.0256$, **Fig 6E**). However, the reductions for SHD on valeric acid ($P > 0.05$, **Fig 6I**) and caproic acid ($P > 0.05$, **Fig 6J**) did not reach significance. In addition, SHD significantly increased the concentration of butyric acid ($P = 0.0266$, **Fig 6G**). There were no differences in fecal isovaleric acid concentration among the four groups (**Fig 6H**).

Similarly, we examined whether the changes in intestinal SCFAs could induce a concomitant change in serum SCFAs. Compared with the Ctrl group, the Model group had significantly higher levels of acetic acid ($P = 0.0009$, **Fig 7D**), butyric acid ($P = 0.0012$, **Fig 7G**), isovaleric acid ($P = 0.0018$, **Fig 7H**), and valeric acid ($P = 0.0009$, **Fig 7I**), and significantly lower level of caproic acid ($P < 0.0001$, **Fig 7J**) level. Nevertheless, the levels of butyric acid ($P = 0.0197$, **Fig 7G**), isovaleric acid ($P = 0.0033$, **Fig 7H**), valeric acid ($P = 0.0023$, **Fig 7I**), and caproic acid ($P = 0.0195$, **Fig 7J**) were restored after SHD supplementation. SHD also significantly reduced isobutyric acid concentration ($P = 0.0079$, **Fig 7F**). There was no statistical significance in propionic acid concentration among the four groups (**Fig 7E**). The results suggested that MCAO-induced SCFAs alternation could be selectively reversed by SHD treatment.

To clarify whether gut microbiota and SCFAs were related to IS, we performed Spearman's correlation analyses and plotted these correlations between cerebral infarct area, gut microbiota abundance, and SCFAs levels as a matrix (**Fig 8A**). The results showed that as the cerebral infarct area increased, the abundance of fecal acetic acid (**Fig 8B**), fecal valeric acid (**Fig 8C**), fecal caproic acid (**Fig 8D**), serum acetic acid (**Fig 8E**), serum valeric acid (**Fig 8F**), *Lactobacillus* (**Fig 8I**), *Desulfovibrio* (**Fig 8J**), and *Bifidobacterium* (**Fig 8K**) also increased.

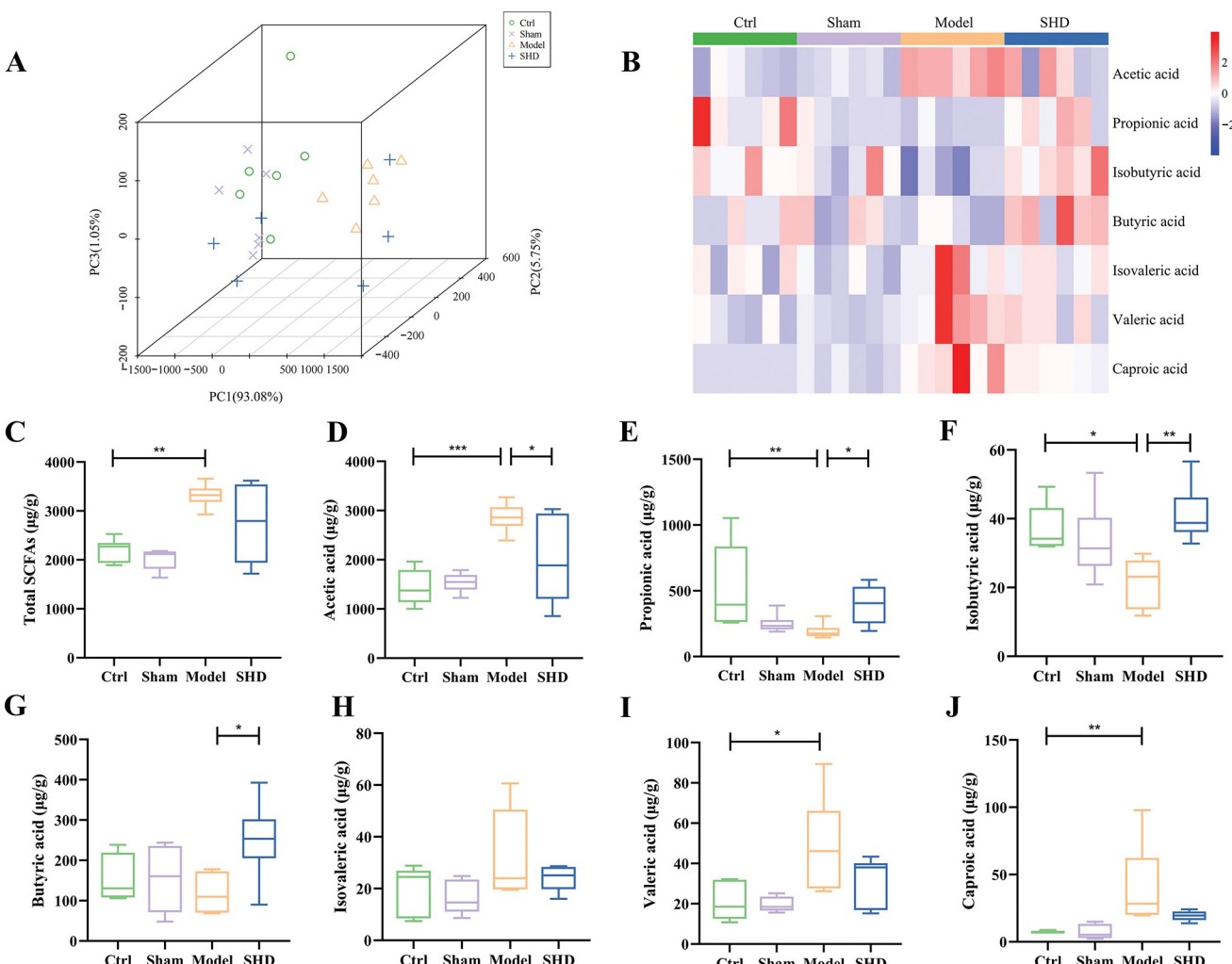

**Fig 6. Fecal SCFA concentrations changed after MCAO modeling and the SHD treatment after stroke restored the SCFAs levels in rats (n = 6).** (A) Three-dimensional principal component analysis plot showing clustering of the fecal SCFA compositions. (B) Heatmap showing the changes in individual fecal SCFAs. (C-J) Concentrations of total SCFAs, acetic acid, propionic acid, isobutyric acid, butyric acid, isovaleric acid, valeric acid, and caproic acid were presented as box plots. Statistical significance was considered at * $P < 0.05$, ** $P < 0.01$, and *** $P < 0.001$.

Meanwhile, the abundance of serum caproic acid (**Fig 8G**) and *Lachnospiraceae* (**Fig 8H**) were significantly negatively correlated with cerebral infarct area, suggesting that SHD could improve the ischemic necrosis of brain tissue in MCAO rats, and the cerebral infarct is correlated with gut microbiota and its metabolites, SCFAs.

## Discussion

SHD is one of the classic prescriptions for IS. It has the effect of harmonizing Qi, blood and body fluids, and relieving the congestion of Fu connecting the environment and internal organs [30]. What's more, it has been reported to exert neuroprotective effects by promoting endogenous neurogenesis and modulating the level of phosphorylated tau after cerebral ischemia/reperfusion injury [6]. In our study, we proved that SHD administration had beneficial effects on IS, which was supported by the pathological morphology results of brain tissue.

The abundance of disordered gut microbiota and SCFAs in MCAO rats was also changed after SHD administration. It has been reported that IS not only causes neurological disorders,

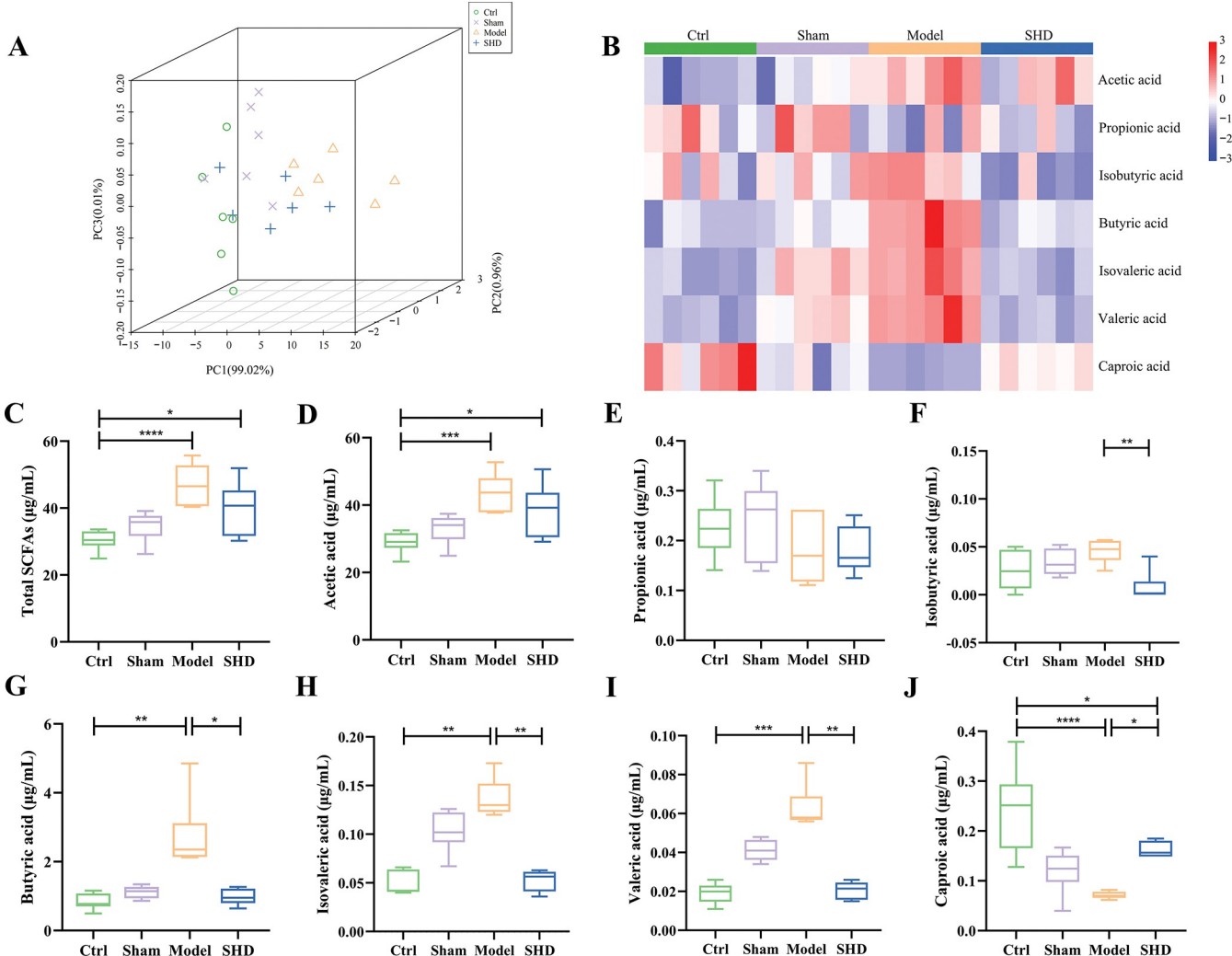

**Fig 7. Serum SCFAs concentrations changed after MCAO modeling and the SHD treatment after stroke recovered the SCFAs levels in rats (n = 6).** (A) Three-dimensional principal component analysis plot showing the serum SCFAs composition among the Ctrl, Sham, Model, and SHD groups. (B) Heatmap represented the changes of individual SCFAs in serum. (C-J) Concentrations of total SCFAs, acetic acid, propionic acid, isobutyric acid, butyric acid, isovaleric acid, valeric acid, and caproic acid of four groups were shown as box diagrams. Statistical significance was considered at * $P < 0.05$, ** $P < 0.01$, *** $P < 0.001$, and **** $P < 0.0001$.

such as the autonomic nervous system and the hypothalamic-pituitary-adrenal (HPA) axis, but also affects the intestinal barrier function by increasing intestinal permeability and altering the intestinal microenvironment [15]. Microbial-derived factors are known to contribute significantly to human health, and changes in microbial community structure can disrupt homeostasis and cause various diseases, including IS [31]. SHD restored the abundances of *Desulfovibrio* (**Fig 4B**, $P < 0.05$), *Lactobacillus* (**Fig 4C**, $P < 0.001$), *Clostridia* (**Fig 4E**, $P < 0.0001$), *Lachnospiraceae* (**Fig 4F**, $P < 0.05$), *Ruminococcaceae* (**Fig 4G**, $P < 0.01$), and *Coprococcus* (**Fig 4H**, $P < 0.05$) in MCAO rats. The aforementioned gut microbiota have been reported to produce SCFAs. Microbes residing in the distal gut influence hypertension, diabetes, multiple sclerosis, affective and cognitive dysfunction, and cardiovascular disease through local synthesis of SCFAs [20, 31]. Gut microbiota-derived SCFAs may directly or indirectly modulate brain functions through immune, endocrine, vagal and other humoral pathways

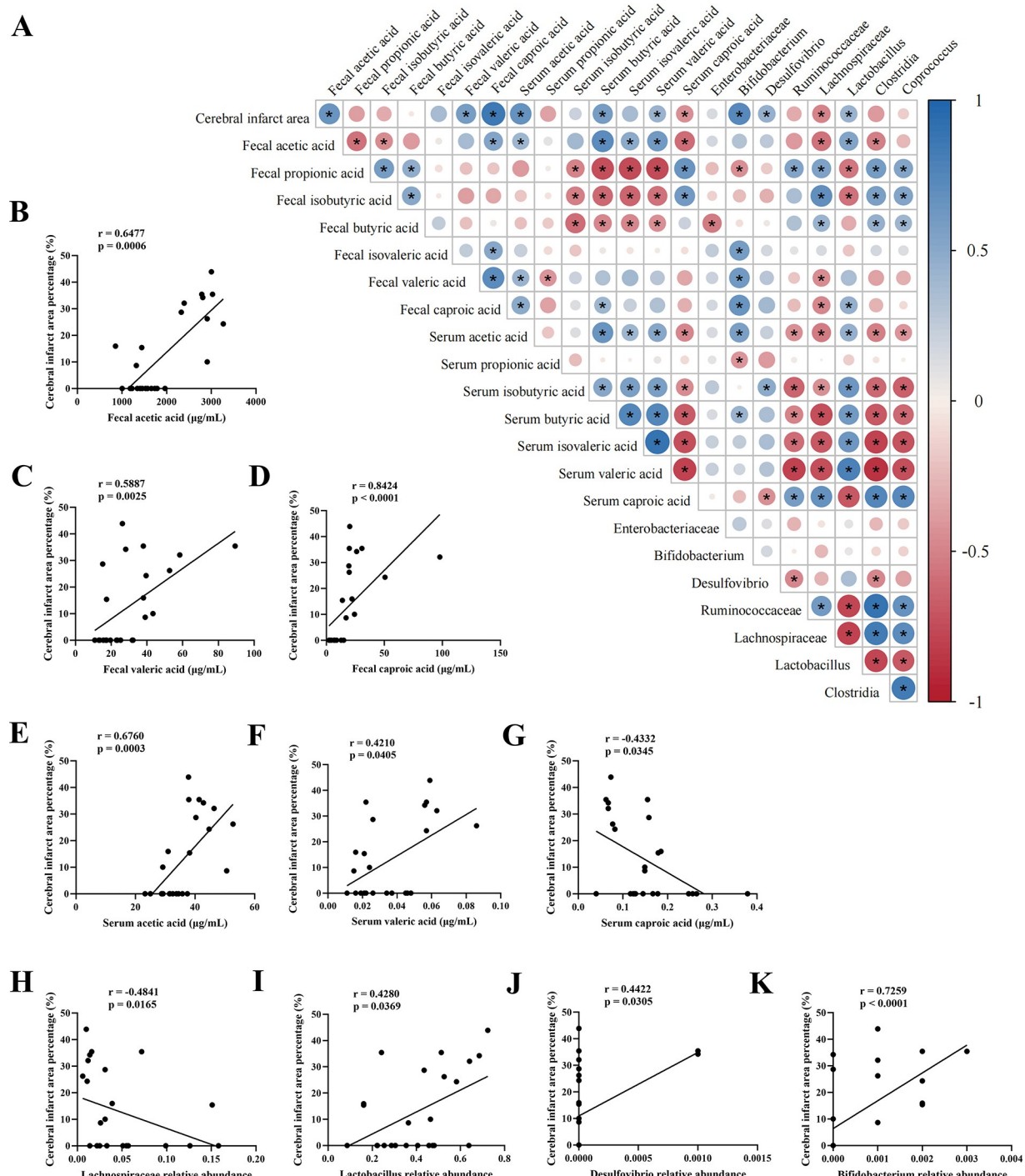

**Fig 8. Spearman's correlation of experimental results.** (A) Heat map of the Spearman's correlations between cerebral infarct area, gut microbial, and SCFAs results. (B-K) Spearman's correlation analysis between cerebral infarct area percentage and fecal acetic acid, fecal valeric acid, fecal caproic acid, serum acetic acid, serum valeric acid, serum caproic acid, *Bifidobacterium*, *Desulfovibrio*, *Lachnospiraceae*, and *Lactobacillus*.

[20], and protect the BBB and intestinal mucosal barrier, whereas lower levels of SCFAs would damage the barriers [32]. Intestinal wall leakage leads to the circulation of pathogenic bacteria and lipopolysaccharide into the bloodstream, exacerbating brain tissue damage across the BBB

[23]. SCFAs are potential therapeutic targets to improve recovery after stroke, as they can activate microglia via circulating lymphocytes to enhance neuronal plasticity [21]. Studies have shown that transplantation of SCFAs-rich fecal microbiota can significantly improve IS recovery, suggesting the critical role of gut microbiota-dependent SCFAs in IS [11, 22]. Therefore, we aimed at the SCFAs-producing bacteria and the changes in SCFAs in MCAO rats to illustrate that SHD could improve ischemic necrosis of brain tissue in MCAO rats accompanied by the reversion of gut microbiota diversity, differential gut microbiota abundance and SCFAs, and further used correlation analysis to examine the relationship between IS, gut microbiota and SCFAs.

Acetic acid, propionic acid, and butyric acid are three major fermentation products produced by the gut microbiota during the breakdown of dietary fiber [33]. A study of feces from IS patients and controls found that IS decreased the concentration of acetic acid (fecal acetic acid concentration in stroke patients was $54.2 \pm 20.3$ μmol/g, and in controls was $66.5 \pm 17.8$ μmol/g, $P = 0.003$), which was negatively correlated with glycated hemoglobin and low-density lipoprotein cholesterol [34]. However, our results showed that acetic acid concentrations in both feces (**Fig 6D**, $P < 0.001$) and serum (**Fig 7D**, $P < 0.001$) were increased in the Model rats and decreased in the SHD group ($P < 0.05$ in feces), probably because brain ischemia can rapidly lead to intestinal ischemia and generate excessive nitrate as a result of free radical reactions, causing intestinal dysbiosis and *Enterobacteriaceae* proliferation [1]. As one of the main manifestations of microbial dysbiosis in IS patients and as an acetic acid producer, *Enterobacteriaceae* may accelerate systemic inflammation and thus aggravate cerebral infarction [1, 15].

Propionic acid is a fermentation product of many bacteria, including *Clostridia* and *Desulfovibrio* [33]. In our result, compared with the Ctrl group, fecal propionic acid was decreased significantly in the Model group (**Fig 6E**, $P < 0.01$), which was restored by SHD administration ($P < 0.05$). It has been reported that propionic acid readily crosses the gut-blood barrier and the BBB. It can be taken up by monocarboxylate receptors in the cerebrovascular endothelium and intestinal lumen, as well as by neurons and glia, where it is thought to be an important energy source in brain metabolism [35]. Propionic acid crosses the BBB to regulate many cell signaling processes, including energy metabolism, neurotransmitter synthesis and release, and lipid metabolism [36].

Butyric acid is the preferred metabolic substrate for intestinal epithelial cells and may enhance intestinal epithelial barrier formation [31]. The mechanism behind the treatment of IS with butyric acid may involve regulation of blood lipids, hemorheology, and histone deacetylase, activation of PPARs, reduction of inflammation, and remodeling of the gut microbiota. These effects lead to beneficial gut microbiota enrichment and intestinal barrier repair [22]. It has been reported that butyrate-producing microbiota remain at low abundance in individuals with IS [27]. In our study, fecal butyric acid (**Fig 6G**, $P < 0.05$) and serum butyric acid (**Fig 7G**, $P < 0.05$) were increased after SHD, and the butyrate-producing microbiota *Lachnospiraceae* (**Fig 4F**, $P < 0.05$), *Ruminococcaceae* (**Fig 4G**, $P < 0.01$), and *Coprococcus* (**Fig 4H**, $P < 0.05$) were increased after SHD. Butyric acid production is regulated by several microbial enzymatic cooperation mechanisms [15]. For example, acetate and butyrate are produced by glycoside hydrolases of *Bacteroides thetaiotaomicron* and *Eubacterium hallii*, respectively [15]. In addition, gut microbiota can synthesize lactate and acetate into butyrate, thereby avoiding lactate accumulation and promoting stabilization of the intestinal environment [37]. The human gut contains both lactate-producing and lactate-utilizing bacteria. High levels of lactic acid have been reported in intestinal disorders [37]. Both *Bifidobacterium* and *Lactobacillus* can produce lactic acid [38]. A significant reduction in *Lactobacillus* was found in a previous study using 16S rDNA sequencing in cynomolgus monkeys undergoing MCAO surgery [39].

Moreover, Jeon et al. investigated changes in gut microbiota composition in a pig MCAO IS model and found that *Lactobacillus* had the highest mean abundance before stroke (33.13 ±5.66%), and its presence reached the lowest point 3 days after stroke (10.63±2.67%); however, it tended to return to prestroke levels at 5 days after stroke (20.19±10.98%) [40]. In our study, *Bifidobacterium* and *Lactobacillus* remained at high levels in the Model rats, and the result was consistent with the levels in high-risk human participants [27]. SHD administration could reduce *Lactobacillus* (**Fig 4C**, $P < 0.001$) but failed to reduce *Bifidobacterium* (**Fig 4D**, $P > 0.05$).

A study of feces from IS patients and controls showed that IS increased valeric acid concentrations (fecal valeric acid concentration in stroke patients was 2.0 ± 1.1 µmol/g, and in controls was 1.3 ± 0.8 µmol/g, $P = 0.003$) [34]. This is consistent with our finding that fecal valeric acid (**Fig 6I**, $P < 0.05$) and serum valeric acid (**Fig 7I**, $P < 0.001$) were significantly higher in MCAO models than in Ctrl. SHD reduced serum valeric acid levels ($P < 0.01$), but there were no statistical differences in fecal levels between the Model and SHD groups ($P > 0.05$). In addition, valeric acid levels were reported to be positively correlated with leukocyte count and high-sensitivity C-reactive protein [34]. In vitro studies showed that propionic acid, valeric acid, and butyric acid inhibited oligomerization of the β-amyloid-(1–40)-peptide. However, only valeric acid inhibited the aggregation of the β-amyloid-(1–42)-peptide [13].

## Conclusion

Here, we presented that symptoms of IS were ameliorated by SHD in MACO rats, as well as the beneficial effects of SHD on gut microbiota and their metabolites, SCFAs. We found that SHD reduced the abnormally elevated *Lactobacillus* and *Desulfovibrio*, but increased *Clostridia*, *Lachnospiraceae*, *Ruminococcaceae*, and *Coprococcus* in MCAO rats. According to KEGG analysis, SHD regulates several pathways, including D-arginine and D-ornithine metabolism, polyketide sugar unit biosynthesis, and cyanoamino acid metabolism in MCAO rats. In addition, SHD reversed elevated acetic acid and decreased propionic acid in the model rats and significantly increased fecal butyric acid. MCAO rats had significantly higher serum levels of acetic acid, butyric acid, isovaleric acid, and valeric acid, and lower levels of caproic acid. Altered serum levels of butyric acid, isovaleric acid, valeric acid, and caproic acid were restored, and the level of isobutyric acid was reduced after SHD administration. We found that cerebral infarct area was positively correlated with fecal acetic acid, fecal valeric acid, fecal caproic acid, serum acetic acid, serum valeric acid, *Lactobacillus*, *Desulfovibrio*, and *Bifidobacterium*; serum caproic acid and *Lachnospiraceae* were significantly negatively correlated with cerebral infarct area. Our result may shed light on the treatment of other models of neuronal injury and, for the first time, provided a direct experimental basis that regulating the composition of the gut microbiota to adjust the levels of SCFAs may be a potential therapeutic mechanism of SHD to clinically ameliorate IS. Further animal and clinical studies are needed to confirm this mechanism.

## Author Contributions

**Data curation:** Yiming Ni, Xiaojun Gou, Wenjie Li.

**Investigation:** Yiming Ni, Xiaojun Gou, Ying Huang.

**Methodology:** Liangyin Cai, Xiaojun Gou, Wenjie Li, Ying Huang.

**Project administration:** Xiaojun Gou, Wenjie Li, Ying Huang.

**Supervision:** Mingmei Zhou, Ying Huang.

**Visualization:** Yiming Ni.

**Writing – original draft:** Yiming Ni.

**Writing – review & editing:** Yiming Ni, Liangyin Cai.

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
