## [Decision Letter · Decision Letter 0]

23 Nov 2023

PONE-D-23-27233Therapeutic Effect of Sanhua Decoction on Rats with Middle Cerebral Artery Occlusion and the Associated Changes in Gut Microbiota and Short-Chain Fatty AcidsPLOS ONE

Dear Dr. Zhou,

Thank you for submitting your manuscript to PLOS ONE. After careful consideration, we feel that it has merit but does not fully meet PLOS ONE’s publication criteria as it currently stands. Therefore, we invite you to submit a revised version of the manuscript that addresses the points raised during the review process.

We look forward to receiving your revised manuscript.

Kind regards,

Muhammad Hassham Hassan Bin Asad, PhD

Academic Editor

PLOS ONE

Additional Editor Comments:

Dear Authors

After careful review it is decided to include minor changes as recommended by the reviewers. Please go through the comments received by the reviewers and submit revised manuscript (with track changing highlighted and other revised manuscript without track changes) along with rebuttal within due date. Thank you for submitting manuscript in the Plos One.

Reviewers' comments:

Reviewer's Responses to Questions

**Comments to the Author**

1. Is the manuscript technically sound, and do the data support the conclusions?

Reviewer #1: Yes

Reviewer #2: Yes

2. Has the statistical analysis been performed appropriately and rigorously? 

Reviewer #1: Yes

Reviewer #2: Yes

3. Have the authors made all data underlying the findings in their manuscript fully available?

Reviewer #1: Yes

Reviewer #2: Yes

4. Is the manuscript presented in an intelligible fashion and written in standard English?

Reviewer #1: Yes

Reviewer #2: Yes

5. Review Comments to the Author

Reviewer #1: The manuscript is well designed and contains a comprehensive information. This covers the all aspects of the title. A few suggestions are recommended to improve the quality and streamline the contents for readers better understanding.

1. Manuscripts require to incorporate some technical literature review on previous and traditional application of Sanhua Decoction (SHD). Although some information is given, but a detail or some reported usage for other disorders.

2. Reference # 33 is cited in several times and this reference how not to be a reason for conlict of intrest. Some major portion of your results are dependendents of the findings of this reference. Pleae provide some logical reasons.

3. In methodology section manuscript contains subheadings from 2.2 - 2-4 and in result section this order and nuber is not followed exactly. however some details information is given but such heading for gene expresssion found in result section.

4. Language check is required especially for discussion section.

Reviewer #2: There are a few observations to be considered before publishing:

All the graphs of results and images are not clear, and the text is too small to be readable.

The word "CTRL" in results from lines 199 and onwards is not mentioned in abbreviations (Page 18).

Other line numbers are: 206, 221, 223, etc.

The discussion is more like a review of the literature and it does not present the comparative studies of any previous data, e.g., the value of polyketide sugar as per literature was nnnn and we observed the improvement as nnn units, etc.

In the discussion, references for comparison are presented, but data is absent in comparison.

In conclusion, no statistical data has been presented and must be included for the scientific community.

6. PLOS authors have the option to publish the peer review history of their article (what does this mean?). If published, this will include your full peer review and any attached files.

Reviewer #1: No

Reviewer #2: No

---

## [Author Response · Author response to Decision Letter 0]

9 Dec 2023

Dear editor

Thank you for taking your valuable time to read the revised manuscript. We have addressed the comments raised by the reviewers, and the amendments are highlighted in the manuscript in revised mode. Below this letter is our explanation on revision according to the comments of the reviewers.

We hope that the revised version of the manuscript is now acceptable for publication in your journal.

I look forward to hearing from you soon.

With best wishes,

Yours sincerely,

Mingmei Zhou, MD, PhD, Associate Professor

Corresponding author

To Reviewer #1：

Thank you for taking your valuable time to read the revised manuscript. The following is our explanation on revision according to the comments.

1. Manuscripts require to incorporate some technical literature review on previous and traditional application of Sanhua Decoction (SHD). Although some information is given, but a detail or some reported usage for other disorders.

Thanks for your suggestions. Sanhua Decoction is a classic Chinese medicine prescription for IS, and we have supplemented its traditional effects in the manuscript (line 63 to 65 and 383 to 384).

2. Reference # 33 is cited in several times and this reference how not to be a reason for conlict of intrest. Some major portion of your results are dependendents of the findings of this reference. Please provide some logical reasons.

Thanks for the comments. Reference # 33 analyzed the concentrations of fecal organic acids in IS patients and control subjects. They found IS was associated with decreased acetic acid and increased valeric acid. 

Our result showed that fecal acetic acid, and valeric acid were significantly increased in MCAO rats, and SHD reversed the changes in acetic acid. In addition, MCAO rats had significantly higher serum levels of acetic acid, and valeric acid, and valeric acid were restored after SHD administration.

Literature #33 examined fecal organic acids in human patients, whereas the aim of our article was to examine SCFAs in MCAO rats and to observe the effect of SHD；and whether there are any differences or similarities with human results for better translation to clinical applications. The difference in acetic acid results may be due to diet, as acetic acid can be ingested through vinegar (apple cider vinegar, rice wine vinegar, etc.). The results for valeric acid were consistent in humans and mice. In addition, we have removed some references to literature # 33 to prevent conflict of interest.

3. In methodology section manuscript contains subheadings from 2.2 - 2-4 and in result section this order and nuber is not followed exactly. however some details information is given but such heading for gene expresssion found in result section.

Thanks for the attentions. We checked and supplemented the content of 16S rRNA gene sequencing (line 173-179).

4. Language check is required especially for discussion section.

Thanks for the attentions. We have reviewed and revised the full language.

To Reviewer #2：

Thank you for taking your valuable time to read the revised manuscript. The following is our explanation on revision according to the comments.

1. All the graphs of results and images are not clear, and the text is too small to be readable.

Thanks for your suggestions. We have uploaded clear images.

2. The word "CTRL" in results from lines 199 and onwards is not mentioned in abbreviations (Page 18). Other line numbers are: 206, 221, 223, etc.

Thanks for the comments. We have added "Ctrl" to the list of abbreviations.

3. The discussion is more like a review of the literature and it does not present the comparative studies of any previous data, e.g., the value of polyketide sugar as per literature was nnnn and we observed the improvement as nnn units, etc. In the discussion, references for comparison are presented, but data is absent in comparison.

Thanks for the attentions. Where possible, we have supplemented the discussion with data for comparison. Regarding as polyketide sugar unit biosynthesis, the pathway was the result of our functional prediction of the gut microbiota, and we did not perform further tests for polyketide sugar. In addition, we deleted the content about the function prediction of gut microbiota in the discussion section.

4. In conclusion, no statistical data has been presented and must be included for the scientific community.

Thanks for your suggestions. We have revised the conclusion section.

---

## [Decision Letter · Decision Letter 1]

21 Jan 2024

Therapeutic effect of Sanhua decoction on rats with middle cerebral artery occlusion and the associated changes in gut microbiota and short-chain fatty acids

PONE-D-23-27233R1

Dear Dr. Zhou,

We’re pleased to inform you that your manuscript has been judged scientifically suitable for publication and will be formally accepted for publication once it meets all outstanding technical requirements.

Kind regards,

Muhammad Hassham Hassan Bin Asad, PhD

Academic Editor

PLOS ONE

Additional Editor Comments (optional):

Reviewers' comments:

Reviewer's Responses to Questions

**Comments to the Author**

1. If the authors have adequately addressed your comments raised in a previous round of review and you feel that this manuscript is now acceptable for publication, you may indicate that here to bypass the “Comments to the Author” section, enter your conflict of interest statement in the “Confidential to Editor” section, and submit your "Accept" recommendation.

Reviewer #1: (No Response)

Reviewer #3: All comments have been addressed

2. Is the manuscript technically sound, and do the data support the conclusions?

Reviewer #1: (No Response)

Reviewer #3: Yes

3. Has the statistical analysis been performed appropriately and rigorously? 

Reviewer #1: (No Response)

Reviewer #3: Yes

4. Have the authors made all data underlying the findings in their manuscript fully available?

Reviewer #1: (No Response)

Reviewer #3: Yes

5. Is the manuscript presented in an intelligible fashion and written in standard English?

Reviewer #1: (No Response)

Reviewer #3: Yes

6. Review Comments to the Author

Reviewer #1: Manuscript is well written and all questions and suggestions are addressed. There is no such remaining queries left to address.

Reviewer #3: I have carefully examined the manuscript, considering its methodology, results, and overall structure. I am pleased to report that the work meets the standards of excellence set by PLOS ONE. The authors have addressed the reviewers observations effectively, and the conclusions drawn are well-supported by the evidence presented. But the graphs of results and images are not still clear, and the text is too small to be readable.

7. PLOS authors have the option to publish the peer review history of their article (what does this mean?). If published, this will include your full peer review and any attached files.

Reviewer #1: No

Reviewer #3: **Yes: **Imran Ahmad

---

## [Editor Report · Acceptance letter]

8 Feb 2024

PONE-D-23-27233R1 

PLOS ONE

Dear Dr. Zhou, 

I'm pleased to inform you that your manuscript has been deemed suitable for publication in PLOS ONE. Congratulations! Your manuscript is now being handed over to our production team.

Kind regards, 

on behalf of

Dr. Muhammad Hassham Hassan Bin Asad 

Academic Editor

PLOS ONE